# Application of observational research methods to real-world studies for rare disease drugs: A scoping review protocol

Yuti P. Patel[1], Lea Ghaddar[1‡], Yuqi Lin[1‡], Nuzat Karim[1‡], Kelvin Chan[2,3], Lee Dupuis[1,4], Mina Tadrous[1*]

1 Leslie Dan Faculty of Pharmacy, University of Toronto, Toronto, Ontario, Canada, 2 Sunnybrook Odette Cancer Centre, University of Toronto, Toronto, Ontario, Canada, 3 Division of Biostatistics, Dalla Lana School of Public Health, University of Toronto, Toronto, Ontario, Canada, 4 The Hospital for Sick Children (SickKids), Toronto, Ontario, Canada

☯ These authors contributed equally to this work.
‡ LG, YL and NK also contributed equally to this work.
* mina.tadrous@utoronto.ca

## Abstract

The primary objective is to identify which observational research methods have been used in the last 5 years in rare disease drug evaluation and how they are applied to generate adequate evidence regarding the real-world effectiveness or safety of rare disease drugs. Rare disease is an umbrella term for a condition which affects <200,000 people each year and despite the rarity of these conditions, collectively they encompass approximately 7000 different conditions. With the striking number of rare conditions, many pharmaceutical manufacturers are introducing an increased number of drugs to treat them. However, due to small patient populations, heterogeneity and other factors related to rare diseases, there are feasibility concerns regarding the generation of adequate efficacy and safety evidence using conventional randomized controlled trials (RCTs). Recently, real-world evidence generated through observational (or real-world) studies has been proposed to address some of the feasibility concerns with RCTs by measuring drug effectiveness or safety in the real-world setting. However, there remain methodological concerns due to a lack of randomization/masking. This proposed scoping review aims to identify which observational research methods in the last 5 years are used in rare disease drug evaluation to address methodological concerns and how they are applied to generate evidence on drug effectiveness or safety. Articles must be primary observational or real-world studies reporting rare disease drug effectiveness or safety published within the five years preceding this review. Literature reviews, meta-analyses, randomized control trials, case series, case reports, opinion pieces, conference abstracts, and studies with unavailable full-text articles will be excluded. The search strategy will combine the following key search concepts: rare disease, drugs for rare disease and observational/real-world studies. The search will be conducted in MEDLINE and EMBASE.

**Review registration number**: Open Science Framework, https://osf.io/f3wpv

**Data availability statement:** No data are associated with this article, as this is a study protocol for a planned scoping review. Data will be collected from publicly available research articles upon completion of the review. All data are in the manuscript and/or supporting information files. Upon completion, extracted data from included studies, a list of excluded studies with reasons, and other supporting materials will be made available as supplementary information in future publications or in a publicly accessible repository. The study protocol has been registered on Open Science Framework: https://osf.io/f3wpv.

**Funding:** The author(s) received no specific funding for this work.

**Competing interests:** The authors have declared that no competing interests exist.

## Introduction

A rare disease, also referred to as an orphan disease, is a medical condition that affects a small proportion of the population. Currently, an international consensus on the definition of a rare disease does not exist. For example, in Canada and the European Union, a rare disease is defined as a condition that affects fewer than 1 in 2,000 people [1,2], while in the United States, it is considered rare if it affects fewer than 200,000 people [3]. By these definitions, approximately 7,000 different conditions qualify as rare diseases [4], resulting in upwards of 300 million people worldwide being affected [5–7]. With a striking number of rare diseases and people affected, the USA and European Commissions implemented the Orphan Drug Act (ODA) in 1983, and Orphan Medicinal Products (OMP) in 2000 to incentivize pharmaceutical manufacturers to develop drugs for rare diseases [1,6,8]. Despite orphan drug incentives being in place for many decades, it is in recent years that an increasing number of drugs for treating rare diseases have been developed. Orphan drug approvals increased from 14 in 2000 to 77 in 2017, and as of recently in 2022, nearly half of all new drug approvals by the FDA were for a rare condition [9].

With respect to drug evaluation, there are data and statistical constraints to generating adequate evidence on the benefits of a therapy for treating a rare disease [4]. From the perspective of drug regulatory and health technology assessment bodies, randomized controlled trials (RCT) are the "gold standard" to evaluate the efficacy and safety of a drug in a particular patient population [10,11]. RCTs are typically designed prospectively, double blinded and randomized, providing equal chance for all patients being allocated to the active treatment or comparator and reduces the extent of bias [11]. The controlled environment of an RCT allows for hypotheses to be tested, though, results from the RCT can only be extrapolated to patients represented in the trial; thus, there are limitations regarding generalizability of results [10]. Additionally, statistical analysis plans for RCTs require an adequate number of patients to statistically power the study [10]. Given the nature of rare diseases, conventional drug evaluations such as RCTs are most often not feasible due to the small patient populations and heterogeneity in the manifestation of the rare disease. These challenges are not unique to rare diseases; however, the limited number of patients amplifies these challenges [4,12].

Given the difficulties in conducting RCT for rare diseases, Real-World Evidence (RWE) has emerged as a viable solution for producing clinical evidence related to the utilization, as well as the potential advantages or risks of a treatment, derived from an analysis of real-world data (RWD) [13,14]. Based on the FDA definition, RWD is "data relating to patient health status and/or the delivery of health care routinely collected from a variety of sources" and can be collected prospectively/retrospectively through observational studies to generate RWE [13,14]. Observational studies such as cohort, case-control, and cross-sectional designs have a well-established history of use in drug research. Observational studies contribute valuable insights into real-world drug effectiveness and safety by capturing data on how medications perform in routine clinical practice. They help elucidate drug utilization patterns, augment the outcomes of RCTs, and complement their findings to provide a more comprehensive understanding of a drug's real-world impact [4,12,13]. In the specific context of rare disease drug evaluation, RWE obtained from observational studies has the potential to address issues associated with limited sample sizes and the generalizability concerns. However, given the absence of randomization and blinding in observational studies, there are methodological concerns regarding the validity of results due to the potential presence of confounding or selection bias [4,12]. While there have been research methods introduced to address these issues, there remains a lack of consistency in their application with respect to rare disease drug evaluation and ultimately the quality of RWE for decision-making. With the widespread adoption of RWE in rare disease drug research, regulatory decision-making, and drug policy over the past 5 years, it is important to understand how established

observational research methodologies have been incorporated to address methodological challenges such as the presence of confounders and small patient populations [10,12,15]. This understanding will aid in informing the appropriate and consistent application of observational research methods to adequately generate RWE for rare disease drugs moving forward. To address this knowledge gap, the proposed scoping review aims to identify in the last 5 years, which observational research methods are being utilized in rare disease RWE drug evaluation and how they are applied to generate adequate evidence regarding the real-world effectiveness or safety.

## Review questions

In the last 5 years, how have observational research methods been used in the generation of RWE on the safety or effectiveness of drugs used to treat rare disease?

a)  Which research methods are being utilized to account for potential confounders or small sample sizes in observational research/studies for rare disease drug research?

b)  How are the research methods identified being applied to generate drug safety or effectiveness in the real-world setting?

c)  Which rare diseases are being studied in observational research/studies in the last 5 years?

## Inclusion criteria

**Participants.** This review will focus on rare diseases, drugs for rare disease, orphan diseases, and orphan disease drugs. Rare disease will be defined in accordance with Health Canada, as a condition that affects fewer than 1 in 2,000 people [2].

**Concept.** This proposed scoping review will consider observational research on rare disease drug effectiveness and/or safety.

**Context.** This review will examine studies using RWD sources such as health administrative data, electronic medical records (EMRs), registry data or databases etc. (all will be documented) from all healthcare settings and regions in the last 5-years.

**Types of sources.** For this scoping review, published observational studies (i.e., cohort, case-control and cross-sectional designs) also referred to as real-world studies, in the last 5-years will be included.

## Exclusion criteria

Studies that do not evaluate an intervention to treat a rare disease based on the definition above; do not use RWD sources; published on or before the five years preceding this review; literature reviews (i.e., systematic, scoping, narrative, etc.), meta-analyses, randomized control trials, case series and case reports, study protocols, opinion pieces (i.e., editorials, commentaries, letters, etc.), conference abstracts, and studies with unavailable full-text articles will be excluded.

## Methods

### Information sources

A structured search will be conducted in MEDLINE and EMBASE. A supplementary search of titles and abstracts of reference lists of included articles will be conducted to identify any additional relevant articles that may have been missed by the search strategy. Reporting will be done in accordance with the Preferred Reporting Items for Systematic Reviews and Meta-Analyses extension for Scoping Reviews (PRISMA-ScR) reporting guidelines [16].

## Search strategy

To initiate the search strategy, a preliminary limited search of MEDLINE was undertaken to identify relevant articles on the topic. The preliminary search indicated the selection of the following key search concepts: rare diseases/ rare disease drugs and observational/real-world studies used in the search strategy. During the preliminary search, it was found that terms like real-world evidence, real-world data, and real-world studies were used interchangeably with established terms like observational studies/observational research. To ensure a comprehensive search study, these concepts were treated as synonyms in this review. Moreover, to ensure the relevance and timeliness of this review, a 5-year time-frame was integrated into this scoping review, considering the recent issuance of RWE guidance documents by various agencies [15]. Upon the selection of the search concepts, the titles and abstracts of relevant articles were scanned to select subject headings and text words used to develop the search strategy for MEDLINE (see S1 Appendix). The subject headings, text words and keyword queries as well as other database specific syntax will be adapted for EMBASE.

## Study records

**Data management.** After conducting searches in both MEDLINE and EMBASE, all identified citations will be transferred to Covidence Systematic Review Software (Veritas Health Innovation in Melbourne, Australia), for screening. To eliminate duplicate citations, we will employ the Bramer de-duplication method [17]. The study selection process will be facilitated through the Covidence Systematic Review Software. Additionally, the data collection process will be managed using Microsoft Excel.

## Selection process

**Title and abstract screening.** To ensure accuracy and consistency in the screening process, two independent reviewers will screen the titles and abstracts of a test set of articles based on the inclusion criteria. Titles and abstracts of articles that do not meet the inclusion criteria will be excluded, and any disagreements or potential modifications to the eligibility criteria will be deliberated by the reviewers. The interrater agreement between the two independent reviewers will be computed using Microsoft Excel and if 80% agreement is obtained between the two reviewers, the remaining articles will be screened independently by the two reviewers. If the agreement percentage falls below 80%, the study team will review and refine the criteria as well as retrain screeners, as necessary [18]. They will then conduct a second round of screening with a new subset of articles until a satisfactory level of agreement is attained. The full-text for articles that potentially meet the criteria will be downloaded and imported into Covidence Systematic Review Software for full-text screening.

**Full-text screening.** The full texts of the articles that successfully passed the screening will undergo a comprehensive evaluation against the inclusion criteria, conducted by two independent reviewers. If any article is found not to meet the inclusion criteria, the rationale for its exclusion will be documented. Any discrepancies between the reviewers will be addressed through discussion and resolved with the input of a third reviewer. The outcomes of the search and study selection process will be outlined in the final scoping review using a flow diagram in accordance with the Preferred Reporting Items for Systematic Reviews and Meta-analyses extension for scoping review (PRISMA-ScR) guidelines [16].

**Data extraction.** Using Microsoft Excel, a data extraction form will be created to record necessary information from all included articles. Prior to the actual data extraction, both reviewers will test the data extraction form on a subset of articles to ensure consistent and accurate data retrieval. Once a consensus is established, both reviewers will independently

extract data from the remaining articles within the same set of included articles. To maintain consistency, a third reviewer will perform random spot checks on 5% of the included articles.

The extracted data variables will include general article characteristics (i.e., year, authors, journal, etc.), study design, rare disease studied, rare disease drug studied and its characteristics (i.e., formulation, drug class), data sources, observational research methods undertaken including the authors' rationale for the choice of methods, outcomes including how they were defined and measured, key findings, conclusions, and funding sources(i.e., industry, academic, government etc). A draft extraction tool is provided (see S2 Appendix). If required, the data extraction tool will be iteratively modified as necessary early during the extraction process, and modification will be detailed in the final scoping review. Any disagreements between the two reviewers will be resolved through discussion with a third reviewer. With respect to missing data, the authors of the paper will be contacted and requested to provide additional information within one week from the date of request. In cases where the data cannot be obtained, the absence of that information will be documented as 'unreported'.

**Data analysis and presentation.** A descriptive quantitative analysis of the included articles will be conducted using Microsoft Excel, alongside a qualitative analysis of appropriate variables as captured in the extraction tool. Data for the first review question will quantify the frequency of each observational research method discussed in the included studies. Data for the second review question will reflect the frequency counts regarding the rationale of method selection and how its application addressed the methodological concerns with the rare disease being studied. Data for the third review question will quantify the frequency of each rare disease discussed in the included studies. The outcomes of the search and the process of including studies will be visually represented in a PRISMA flow diagram as per reference [16]. The relevant data gathered during the extraction will be illustrated in suitable tables and figures, serving to highlight the current evidence and knowledge gaps in literature. A descriptive summary of how observational research methods included in the review were applied to generate the health outcomes of interest will be included with the tabular results. The findings from this scoping review will support researchers in understanding which observational research methods can be applied to certain rare disease drug studies to overcome methodological challenges to obtain RWE on drug effectiveness or safety.

## Supporting information

**S1 Appendix. Search strategy.**
(DOCX)

**S2 Appendix. Data extraction form.**
(DOCX)

**S3 Appendix. PRISMA scoping review checklist.**
(DOCX)

## Author contributions

**Conceptualization:** Yuti P. Patel, Kelvin Chan, Lee Dupuis, Mina Tadrous.

**Data curation:** Yuti P. Patel, Mina Tadrous.

**Formal analysis:** Yuti P. Patel, Lea Ghaddar, Yuqi Lin, Nuzat Karim, Mina Tadrous.

**Funding acquisition:** Yuti P. Patel, Mina Tadrous.

**Investigation:** Yuti P. Patel, Mina Tadrous.

**Methodology:** Yuti P. Patel, Kelvin Chan, Lee Dupuis, Mina Tadrous.

**Project administration:** Yuti P. Patel, Mina Tadrous.

**Resources:** Yuti P. Patel, Mina Tadrous.

**Software:** Yuti P. Patel, Mina Tadrous.

**Supervision:** Kelvin Chan, Lee Dupuis, Mina Tadrous.

**Validation:** Lea Ghaddar, Yuqi Lin, Nuzat Karim, Kelvin Chan, Lee Dupuis, Mina Tadrous.

**Writing – original draft:** Yuti P. Patel.

**Writing – review & editing:** Yuti P. Patel, Lea Ghaddar, Yuqi Lin, Nuzat Karim, Kelvin Chan, Lee Dupuis, Mina Tadrous.

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
