## [Editor Report · Decision Letter 0]

18 Jun 2024

PONE-D-24-19336

Application of observational research methods to real-world studies for rare disease drugs: a scoping review protocol

PLOS ONE

Dear Dr. Patel,

Thank you for submitting your manuscript to PLOS ONE. After careful consideration, we have decided that your manuscript does not meet our criteria for publication and must therefore be rejected.

The protocol is not within the scope of the journal. PLOS ONE only considers protocols for systematic reviews and meta-analyses.

I am sorry that we cannot be more positive on this occasion, but hope that you appreciate the reasons for this decision.

Kind regards,

Tim Mathes

Academic Editor

PLOS ONE

- - - - -

---

## [Author Response · Author response to Decision Letter 1]

23 Oct 2024

This manuscript has been resubmitted following a successful appeal for reconsideration. Please refer to the following appeal number: PONE-D-24-19336 and email exchange with Melanie Española.

---

## [Decision Letter · Decision Letter 1]

14 Jan 2025

PONE-D-24-19336R1Application of observational research methods to real-world studies for rare disease treatments: a scoping review protocolPLOS ONE

Dear Dr. Patel,

Thank you for submitting your manuscript to PLOS ONE. After careful consideration, we feel that it has merit but does not fully meet PLOS ONE’s publication criteria as it currently stands. Therefore, we invite you to submit a revised version of the manuscript that addresses the points raised during the review process.

We look forward to receiving your revised manuscript.

Kind regards,

Kiyan Heybati

Academic Editor

PLOS ONE

Journal Requirements:

4. As required by our policy on Data Availability, please ensure your manuscript or supplementary information includes the following:

5. We note that your manuscript is not formatted using one of PLOS ONE’s accepted file types. Please reattach your manuscript as one of the following file types: .doc, .docx, .rtf, or .tex (accompanied by a .pdf).

If your submission was prepared in LaTex, please submit your manuscript file in PDF format and attach your .tex file as “other.

6. Please amend the title either on the online submission form or in your so that they are identical.

"Application of observational research methods to real-world studies for rare disease treatments: a scoping review protocol

Reviewers' comments:

Reviewer's Responses to Questions

**Comments to the Author**

1. Does the manuscript provide a valid rationale for the proposed study, with clearly identified and justified research questions?

Reviewer #1: Yes

Reviewer #2: Partly

2. Is the protocol technically sound and planned in a manner that will lead to a meaningful outcome and allow testing the stated hypotheses?

Reviewer #1: Yes

Reviewer #2: Partly

3. Is the methodology feasible and described in sufficient detail to allow the work to be replicable?

Reviewer #1: Yes

Reviewer #2: No

4. Have the authors described where all data underlying the findings will be made available when the study is complete?

Reviewer #1: Yes

Reviewer #2: Yes

5. Is the manuscript presented in an intelligible fashion and written in standard English?

Reviewer #1: Yes

Reviewer #2: Yes

6. Review Comments to the Author

You may also provide optional suggestions and comments to authors that they might find helpful in planning their study.

Reviewer #1: In the study protocol “Application of observational research methods to real-world studies for rare disease

treatments: a scoping review protocol,” the authors outline a methodology to identify observational research methods used over the past 5 years and evaluate their application. This work is relevant to those closely related to their field.

Minor revision suggestions:

• In the introduction, please place references 1 – 4 with the respective piece of information rather than all together at the end.

• Under exclusion criteria the authors state English as a language restriction. A scoping review should not technically have a language restriction. If this is appropriate in this case, it should be justified.

• Under exclusion criteria the authors mention that case reports and series will be excluded, however these are often also observational in nature; please clarify.

• Consider adding the use of a medical librarian to assist in setting up key words and concepts.

Reviewer #2: Title: Looks interesting and identifies the report as a scoping review protocol

Short title: Observational research methods for rare disease treatments (suggestion)

Abstract:

7000 different conditions looks like a huge number of diseases, are the authors looking at any specific diseases (it might be good to keep the diseases to one condition or specialty) for better results (suggestion).

Can the authors change their search duration till 2024?

Languages?

Will they consider implementation studies?

MEDLINE and EMBASE looks very narrow for this research question? Please consider adding subject specific databases, if none exist – please mention

Some terms like – observational research methods, rare disease would be helpful for readers and standardizing the terminology.

Competing interests:

Looks like MT has some conflicts of interests reported in a paper, this can be mentioned here in this paper as well?

“M. T. received financial support from the Canadian Agency for Drugs and Technologies in Health. M. T., T. A., and K. N. H. report a relationship with Canadian Agency for Drugs and Technologies in Health that includes consulting or advisory.”

(ref: Development of a Canadian Guidance for reporting real-world evidence for regulatory and health-technology assessment (HTA) decision-making., https://doi.org/10.1016/j.jclinepi.2024.111545. Journal pre proof)

If there are concerns with conventional RCTs it might be helpful to look at concerns related to efficacy, safety evidence too?

Can they address concerns with real world studies as well? They aren’t planning to look at RCTs.

Introduction:

Nice introduction.

It might be helpful to mention how observational studies may contribute to effectiveness in real world drug (line 97,98)

Can a reference be added for statement 107,108 please?

Imperative sounds too strong – can this very be changed to ‘might be helpful’. If its imperative, might as well perform RCTs?

Rare diseases – looks very broad, can the authors consider narrowing the participants?

Can they also explain and add some detail on the existing research methods for better understanding. And expand on the ‘information sources’ please. Add a note on grey literature search as well. Can they also add a note on how they will conduct their supplementary search (line 151-153) especially in relation to relevant literature searches.

Data management: How they will manage data using Excel is unclear, if they could explain which data will be managed it might be helpful.

The inclusion criteria need to be bit more robust and detailed to prevent any future concerns during the screening phase- hope the authors will consider the suggestion to make the criteria a bit more detailed.

Regarding missing data – line 219 to 220, is there are a time frame the authors will wait while they contact the study authors, any time frames for sending reminders?

Will they use Covidence for data extraction as well? Might be good to mention it here.

Data analysis and presentation:

It might be good to mention the type of quantitative and qualitative analysis that will be performed please?

Inclusion criteria: please explain the concept in detail to prevent ambiguity, like conditions or please consider adding an appendix

Context: the term ‘health administrative data’ sounds confusing please explain.

I am not sure how the authors are planning to search unpublished studies? Please expand the types of sources.

Others: Please consider updating the PRISMA ScR checklist based on the revision 1 protocol.

A few questions:

Not sure how they might be able to generate evidence on drug effectiveness or safety?

It might be good to explain real world effectiveness methods

Rare disease drugs and evaluation.

7. PLOS authors have the option to publish the peer review history of their article (what does this mean? ). If published, this will include your full peer review and any attached files.

**Do you want your identity to be public for this peer review?** For information about this choice, including consent withdrawal, please see our Privacy Policy .

Reviewer #1: No

Reviewer #2: No

---

## [Author Response · Author response to Decision Letter 2]

23 Feb 2025

Hi,

Thank you for the comments from the editors. They were very insightful. I've provided responses to each comment and look forward to this protocol being published.

Yuti

---

## [Editor Report · Decision Letter 2]

25 Feb 2025

Application of observational research methods to real-world studies for rare disease drugs: a scoping review protocol

PONE-D-24-19336R2

Dear Dr. Patel,

We’re pleased to inform you that your manuscript has been judged scientifically suitable for publication and will be formally accepted for publication once it meets all outstanding technical requirements.

Kind regards,

Kiyan Heybati

Academic Editor

PLOS ONE
---

## [Editor Report · Acceptance letter]

PONE-D-24-19336R2

PLOS ONE

Dear Dr. Patel,

I'm pleased to inform you that your manuscript has been deemed suitable for publication in PLOS ONE. Congratulations! Your manuscript is now being handed over to our production team.

Kind regards,

on behalf of

Dr. Kiyan Heybati

Academic Editor

PLOS ONE